# How Can the Digital Economy Promote the Integration of Rural Industries—Taking China as an Example

**Zepu Zhang, Chen Sun \* and Jing Wang**

College of Economics and Management, Shanghai Ocean University, Shanghai 201306, China;
18638797164@163.com (Z.Z.); jwangocean@163.com (J.W.)
\* Correspondence: chensun@shou.edu.cn; Tel.: +86-156-9216-5636

**Abstract:** The rapid development of China's digital economy has promoted the digital transformation of Chinese society and become a new driving force for China's social development. Furthermore, increasing farmers' income is the central task of China's "three rural issues" project, and the integrated development of primary, secondary, and tertiary rural industries is an important way to revitalize rural industries and broaden farmers' income channels, so it is very important to promote the integrated development of rural industries. In the context of the rapid development of China's digital economy, it is of great significance to study how the digital economy promotes the integration of rural industries. Therefore, this study analyzed how China's digital economy promotes the integration of rural industries by using the panel data and regression models of 30 provinces (municipalities and districts) in China from 2011 to 2021. The results show that (1) there is a significant positive relationship between the development level of the digital economy and the level of rural–industrial integration and that the development of the digital economy promotes the process of rural–industrial integration; (2) intermediary effect analysis shows that as the digital economy promotes rural–industrial integration, scientific and technological innovation levels and rural human capital are important intermediary channels; and (3) spatial analysis shows that the development of the digital economy can not only promote the integration of rural industries in the province but also have a positive spatial spillover effect on neighboring provinces. Therefore, it is necessary to create a good atmosphere for the development of the digital economy, constantly promote the development of the digital economy, pay attention to scientific and technological innovation and rural talent training, and promote the integration and coordinated development of the digital economy and rural industries between regions in order to improve the level of rural–industrial integration and contribute to the rural revitalization strategy.

**Keywords:** digital economy; rural–industrial integration; mediating effect; spatial spillover effect





## 1. Introduction

At present, China has eliminated absolute poverty, completed the task of poverty alleviation, and is moving towards its second centenary goal. In recent years, China has gradually entered the era of the digital economy, and the digital productivity spawned by digital technologies such as the Internet, blockchain, artificial intelligence, and big data is promoting the transformation of social production relations and bringing huge development opportunities. The rapid development and popularization of China's digital economy has profoundly changed human life and social production modes, even affecting the future direction of China's economy and society. At the same time, in the process of deepening its rural revitalization strategy, China has continuously improved rural industries' levels of integration and development, which has played an important role in broadening channels for farmers to increase their employment and income and accelerating the modernization of China's agriculture and rural areas. From this point of view, it is very important to actively guide and develop China's rural–industrial integration to improve

the conditions of China's rural areas. However, there are many factors affecting China's rural–industrial integration. For instance, how does the digital economy affect China's rural–industrial integration in the context of social development, and what is its influence mechanism and effect? This paper discusses and analyzes these problems in depth by using relevant data on Chinese provinces and economic regression models. Studying how the digital economy affects rural–industrial integration is of great significance in actively guiding the development of the digital economy, as it will aid the discovery and use of the best paths and channels for promoting rural–industrial integration and China's rural development.

Tapscott [1], an American scholar, first proposed the term and described 12 significant features of the "digital economy", but he did not fully define it. Later, relevant scholars provided different definitions of the digital economy. The G20 Digital Economy Development and Cooperation Initiative defines the digital economy as an economy in which the key production factors are digital knowledge and information, the main carrier is the modern information network, and information and communication technology are used to improve efficiency and optimize economic structures [2]. Academia has carried out significant research on the digital economy, obtaining abundant data and reaching important conclusions. Relevant international studies have found that the digital economy has significant positive effects on national economic growth [3], industrial transformation and upgrading [4], carbon emission reductions [5], social governance system improvements [6], the development of developing countries [7], and air pollution reductions [8]. Chinese scholars have mainly studied three aspects of the digital economy. The first is its measurement. Some scholars have measured the scale of the digital economy based on the added value of core industries [9], built a comprehensive evaluation index system based on different economic form attributes to measure the digital economy's development level [10], and used the input–output data of digital economy industries to measure the regional efficiency of digital economy industries [11]. Some scholars have also measured the development level of the digital economy through comparisons between different nations [12], provinces [13,14], and agriculture and rural areas [15]. The second aspect is qualitative analysis. These researchers argue that to promote the sustainable and healthy development of the digital economy, it is necessary to properly understand the relationship between the market and the government [16]. Some scholars have also discussed the internal logic and realization paths of rural revitalization [17] and common prosperity [18] enabled by the digital economy, while others have researched the development mechanisms and paths of new rural digital economy formats [19], how to build a digital economy governance system with Chinese characteristics [20], and the realization paths of driving green consumption development [21]. The third aspect is quantitative analysis. Here, most scholars have focused on digital economy development and trade [22,23], export [24], human capital [25], resident income [26] consumption [27], economic growth [28–30], industrial development [31], and green and low-carbon development [32,33].

The concept of rural–industrial integration originated from the "Sixth Industrialization" concept put forward by Japanese scholar Nara Imamura in the 1990s, which marked a breakthrough in the development of rural industry [34]. However, China's history of the exploration and development of the integration of rural industries began before that. The first stage was the agribusiness-integrated management stage (1978 to the early 1990s). This stage can also be divided into two periods. The first half of the period was the agribusiness complex (1978 to mid-1980s). In September 1978, China began to learn from the Yugoslav Bekebe model and implemented pilot agro–industrial and commercial joint enterprises. After several years of pilot programs and promotion, the establishment of agricultural and industrial joint enterprises gradually cooled down until disappearing after 1983. The second half of the period was the agribusiness complex (1983 to the early 1990s). In 1983, the No. 1 Document of the Central Committee of the People's Republic of China proposed that only by engaging in the all-round development of agriculture, forestry, animal husbandry, sideline, and fishery in tandem with the comprehensive management of agriculture,

industry, and commerce in rural China can the virtuous cycle of agricultural ecology be maintained and economic benefits improved. With the drastic economic and social changes in the middle and late 1980s and the rapid development of township enterprises, calls for the comprehensive management of agriculture and commerce gradually fell silent and faded away in the early 1990s. The second stage has been the agricultural industrialization management stage (since 1993). In this stage, the development experience of agricultural industrialization can be divided into three periods. The first period comprised the initial period of exploration (1993 to 2001). The primary aims of this period were to adapt to the socialist market economic system that China had just established and to solve problems in the connections between the production, processing, and marketing of agricultural products. The second period was the big and strong period (2001 to the early 2010s). After China's accession to the World Trade Organization in 2001, China significantly opened itself to the outside world, and its agriculture began to face direct competition from international agriculture industries. Accordingly, it became urgently important to improve agricultural competitiveness. After development, China's leading agricultural industrialization enterprises formed more mature development models, such as "leading enterprises and farmers", "market and farmers", and "scale characteristics industry and farmers". The third period has been the period of deep integration (early 2010s to the present), in which agricultural industrialization has been promoted as part of the integration of rural industries. In 2015, China's "Central No. 1" document first proposed the development strategy of the "Rural integration of one, two and three industries" (referred to as "rural industry integration"). Since then, the strategy has become the guideline used to solve the problem of "three farmers" and promote agricultural modernization. The fundamental purpose of the strategy is to extend the agricultural industry into the secondary and tertiary sectors, promote the integration and interaction of agriculture with industry, make agriculture into a comprehensive industry, improve the agricultural industry chain, increase the added value of agricultural products, expand the sales channels of agricultural products, and increase farmers' income [35]. Research on rural–industrial integration has mainly focused on three aspects. The first aspect is measuring the level of rural–industrial integration. To construct a comprehensive evaluation index system for the integrated development of rural industries, researchers must measure the level of the integrated development of rural industries in China [36] and study the spatial distribution characteristics [37] and development quality of this integrated development [38]. The second aspect comprises the factors affecting the integration of rural industries. Developing digital inclusive finance [39–41], improving the levels of innovation in agricultural science and technology [42], expanding the scale of land management [43], developing agricultural cooperatives [44], improving levels of urbanization [45], and increasing fiscal support for agriculture [46] are all conducive to promoting the integration of rural industries. The third aspect is the social effect caused by the integration of rural industries. The integrated development of rural industries plays an important role in promoting regional economic development [47], promoting green agricultural development [48], raising farmers' income [49], narrowing the income gap between urban and rural areas [50], improving farmers' quality of life [51], alleviating the multidimensional poverty of rural households [52], and promoting rural revitalization [53].

Regarding the relationship between the digital economy and the integration of rural industries, relevant studies have shown that the digital economy can optimize the input structure of factors through the substitution effect, improve the production synergy and innovation of various sectors through penetrative and destructive effects [54], break the boundary restrictions of traditional industries and technologies, and promote industrial and technological integration [55]. In recent years, the rapid development of e-commerce for agricultural products in China has, to a large extent, performed the link-and-matching function of the digital economy, driving the accelerated integration of agricultural and industrial services [56]. According to transaction cost theory, the digital economy uses information as an important factor of production that can reduce production costs, improve production efficiency, promote the transaction of agricultural means of production, and

provide favorable conditions for the digital economy to penetrate and promote the integration of rural industries [57]. As an important part of the digital economy, digital finance provides financial support for various entrepreneurial projects and production inputs of agricultural business entities as well as facilitating the integrated development of multiple agricultural functions [58]. In farming, for example, digital finance not only expands the production function of agriculture but also increases the leisure function of agriculture, which signifies the good performance of the integration of agriculture and service industries [59]. Chen Yiming argued that the digital economy and rural industry can also be integrated, and technological innovation plays an important role in this integration process [60]. On the contrary, the current supply-side shortage of China's digital agricultural construction, the uneven development level of the information infrastructure in different regions, and the lagging construction of data-sharing systems have seriously hindered the integrated development of rural industries and limited the driving force of the digital economy on rural–industrial integration [61]. In the Chinese government, government departments lack colleagues who understand information technology, have management skills, and have rich experience in rural work [62], so they have not engaged in top-level design and policy formulation, which is a difficult problem affecting the promotion of rural–industrial integration in the digital economy. In short, most studies agree that the development of the digital economy can promote the integration of rural industries, though a small number of scholars have expressed concerns.

Researchers have carried out many studies on the concept definition, connotation interpretation, impact effects, and implementation paths of the integration of the digital economy and rural industries, but few studies have been conducted from an empirical perspective. Accordingly, the possible contributions of this paper are as follows. First, by combing the relevant literature and describing how the digital economy promotes the integration of rural industries, this paper verifies the significant promoting effect of the development of the digital economy on the integration of rural industries through empirical analysis. Second, in terms of the path mechanisms of the digital economy used to promote rural–industrial integration, this study verifies and supports the positive intermediary roles of scientific and technological innovation levels and rural human capital, which enriches existing research. Third, in this study, we analyzed the spatial aggregation of the digital economy and rural–industrial integration from the provincial level in China, thus verifying the spatial spillover effect of the digital economy on rural–industrial integration.

## 2. Theoretical Analysis and Research Hypothesis

Digital economy affects rural industrial integration through five aspects. First, the digital economy promotes the integration of rural industries by extending the agricultural industry chain. The combination of digital technology and agriculture gives agriculture intelligent functions such as monitoring. It can not only monitor the growth process and growth environment of agricultural products in real time but also master the data of the agricultural factor input and agricultural product sales so as to make the information exchange between the upper, middle, and lower reaches smoother and solve the problem of information asymmetry. At the same time, consumers can also monitor the production, processing, and circulation of agricultural products through the monitoring system, and it is also convenient for consumers to trace the quality of agricultural products [63], which improves consumers' willingness to consume and drives the effective supply of agricultural products. Therefore, the digital economy drives the integration of agricultural production, processing, circulation, sales, and services through information transparency as well as stimulates consumption and supply, thus promoting the integration of rural industries. Second, the digital economy promotes the integration of rural industries by expanding the versatility of agriculture. Farmers have enhanced their connection with the market through digital technology and can formulate production and sales plans according to local conditions [64] so that more agricultural products can enter the market and increase the sales volume and added value of agricultural products. Digital technology can also combine

agricultural products with local cultural and geographical advantages, create unique brands, and promote the development of characteristic agriculture, leisure agriculture, and cultural tourism through media such as short videos. In addition, the digital and precise management of agriculture can discover and reduce the input of agricultural products and reduce the damage to the rural ecological environment [65]. Therefore, the digital economy promotes the integration of rural industries by developing the economic, cultural, and ecological functions of agriculture. Third, the digital economy promotes the integration of rural industries through the development of new agricultural formats. In the process of combining digital technology with agriculture, the involvement of the substitution effect, penetration effect, and synergistic effect can break the industry barriers of rural primary, secondary, and tertiary industries; sell agricultural products through e-commerce platforms; promote the integration of production and marketing; and promote the development of new business forms such as order agriculture. At the same time, the digital economy promotes the rise of new formats such as rural tourism, boutique homestays, wellness, and farmhouses through digital technologies such as big data, blockchain, and cloud computing, thereby promoting rural–industrial integration. Fourth, the digital economy promotes the integration of rural industries by realizing the integrated development of agricultural service industries. The widespread use of digital payment in rural tourist attractions, rural homestays, farmhouses, rural supermarkets, and other places has effectively catered to the online payment habits of urban residents, enhanced the comfort and satisfaction of urban residents in rural tourism and consumption, and promoted the development of rural tourism. At the same time, digital finance can effectively reduce the threshold of financial services and expand the supply of rural finance and broaden the access to financial services for agricultural business entities, thus promoting the integration of agricultural service industries. Fifth, the digital economy promotes the integration of rural industries by improving the interest-linkage mechanism. Rural e-commerce can use digital technology to promote and sell agricultural products, increase farmers' income, and also increase the income of e-commerce platforms, packaging and processing enterprises, logistics, and other industries. At the same time, digital technology can timely and accurately transmit data such as consumer demand and preference to all aspects of agricultural product production and service; guide rural industrial operators to produce, process, and circulate according to consumer demand; and build an agricultural product quality traceability system with "information chain-evidence chain-trust chain" as the main line, making it possible to accurately track agricultural products [66] so as to form a benign interaction between farmers and enterprises in sharing rights and interests and sharing risks in order to strengthen cooperation and form an interest-linkage mechanism to promote rural industrial integration. Accordingly, we put forward

**Hypothesis 1.** *The development of digital economy helps to promote the integration of rural industries.*

Innovation is a process of establishing a new production function, creating new value by introducing new production factors and combinations into a production system [67]. The digital economy affects the flow of innovative knowledge through various channels, such as increasing knowledge stock, improving the speed of information transmission, and reducing information asymmetry [68]; integrating and aggregating innovative knowledge resources and reducing enterprise innovation costs [69]; breaking space–time constraints and changing industrial models [70]; and optimizing the allocation of scientific and techno-logical innovation resources [71], thus providing a good atmosphere and foundation for the development of innovative activities [72] and promoting scientific and technological innovation. At the same time, the development of the digital economy leads to the further opening of the organizational model of innovation entities. In the process of developing and designing new products or services, external entities can be more easily involved in certain forms, and the information barrier between developers and demanders will be reduced, thus improving bridging degrees and reducing innovation risks [73]. The devel-

opment and popularization of digital technology has improved society's ability to accept new things, forcing researchers to engage with stronger innovative interests and ideas for new things and providing favorable conditions for innovation in terms of atmosphere and environment [74]. The digital economy has dispersal functions such as interconnection, spillover, and diffusion, that is, the "dandelion effect", which can improve the level and efficiency of collaborative innovation among various regions [75]. In the process of economic and social development, scientific and technological innovation is the main driving force that promotes the integration of the three industries [76]. Once an effective scientific and technological innovation appears in an industry, resulting in many benefits for related enterprises, this technology spreads and penetrates the industry at a rapid speed, which is the positive externality referred to in economics. Industrial integration is the product of science, technology, and system innovation, and innovation is one of the key driving forces in the promotion of industrial integration [77–79]. Scientific and technological innovation changes the agricultural production mode by changing the function, form, and quality of agricultural products, leading to improvements in the quality level and development speed of the agricultural industry and creating good conditions for the integration of agriculture with the secondary and tertiary industries [76]. The application of new technologies in agricultural production is conducive to the realization of intelligent, digital, and information technology in agricultural production, breaking the technical barriers between various departments within agriculture and between agriculture and the second and third industries, changing the production characteristics and value creation process of agricultural products, and enabling to the integration of rural industries [80]. For example, the application of internet information technology, Internet of Things technology, credit payment technology, and warehousing and logistics technology in agricultural production and operation has made agriculture deeply integrated with e-commerce, modern logistics, financial lending, and other secondary and tertiary industries. To sum up, the development of the digital economy provides a good atmosphere and channels for scientific and technological innovation, penetrating agriculture through positive externalities and therefore promoting the integration of rural industries. Accordingly, we put forward

**Hypothesis 2.** *The digital economy promotes the integration of rural industries by improving the levels of scientific and technological innovation.*

The prosperity of talent is a key factor in the integrated development of rural industries [81]. The rapid development of the digital economy has improved farmers' access to information and knowledge and reduced farmers' information collection costs and learning costs. The core of the digital economy is digital technology, which is mainly characterized by difficulty in innovation, strong periodicity, and fast upgrading and iteration [26]. The digital economy will cultivate farmers' crisis awareness and force them to learn new knowledge and skills to meet the needs of digital technology, resulting in the gradual improvement of rural human capital. At the same time, digital technology makes all kinds of learning resources easy to obtain. In addition to promoting basic knowledge on subjects such as Chinese and mathematics, digital technology also promotes the dissemination of professional agricultural knowledge in rural areas. Farmers can obtain effective information resources and scientific knowledge, and then, they can update their ways of thinking and knowledge systems. Accordingly, subjective ability to adapt to the development of rural e-commerce and agricultural information should be given full play in network marketing models. Human capital is a powerful driving force in the promotion of the integration of rural industries [47] and plays a key role in the process of agricultural development under the "production–management–industry" model. If the level of rural human capital in a region is low, the local rural labor force has a relatively low cultural quality, weak awareness of Internet use, and weak ability to apply digital technology, which is not conducive to the rural labor force's obtainment of information regarding the development of agricultural industry on digital platforms, and it becomes more difficult to use agricultural digital

technology to deeply integrate with other industries. This situation is not conducive to promoting the integrated development of rural industries. On the contrary, if the level of rural human capital in a region is high, the rural labor force as a whole has a higher cultural quality, which is conducive to the efficient use of agricultural digital technology and the collection of agricultural resource information, which is conducive to the integrated development of rural industries [82]. The development of the digital economy will enable farmers to obtain advanced, effective, and low-cost information and knowledge; gradually acquire modern skills and concepts; and enhance rural human capital, thus promoting the integrated development of rural industries. To sum up, the digital economy can improve rural human capital by reducing farmers' learning costs, expanding learning channels, and improving farmers' subjective initiative in learning, allowing them to better grasp modern knowledge and promote rural–industrial integration. Accordingly, we put forward

**Hypothesis 3.** *The digital economy can promote rural–industrial integration by improving rural human capital.*

Based on the above analysis and the proposed Hypotheses 1, 2, and 3, the relationship between the digital economy and scientific and technological innovation levels, rural human capital, and rural–industrial integration can be mapped, as shown in Figure 1.

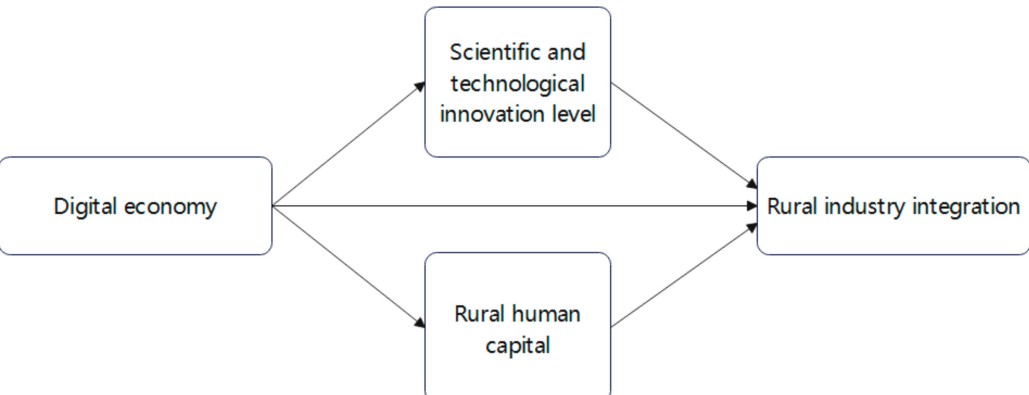

**Figure 1.** Relationship between the digital economy, rural–industrial integration, and intermediary variables. Note: Figure 1 is based on the theoretical analysis and research hypothesis in Section 2, from which the authors draw the relationship diagram of four variables.

## 3. Materials and Methods

### 3.1. Materials

3.1.1. Data Source

This paper selected the panel data of 30 provinces (municipalities and districts) in China from 2011 to 2021 (except Tibet, Hong Kong, Macao, and Taiwan). The data mainly came from the official websites of the National Bureau of Statistics, China Statistical Yearbook, China Rural Statistical Yearbook, Peking University Digital Inclusive Finance Development Index Report, China Leisure Agriculture Yearbook, China Agricultural Product Processing Industry Development Report, National Greenhouse Data System, and China Population and Employment Statistical Yearbook. Some data gaps were filled in with the linear interpolation method.

3.1.2. Variable Selection

The level of rural–industrial integration was selected as the explained variable. In terms of rural–industrial integration indicators, this study referred to the research of Lai [83], Wang [84], Zhang [85], Hao [86], Zhang [87], and Zhang [88] and selected five indicators: the extension of the agricultural industry chain, expansion of agricultural

versatility, cultivation of new forms of agriculture, integration of agricultural services, and an improved interest-linkage mechanism. The extension of the agricultural industry chain is the basic manifestation of the integrated development of rural industries, and it mainly occurs through the deep processing of primary agricultural products, including production, processing, and sales, to increase their added value. The extension of agricultural versatility refers to the use of natural, ecological, and cultural resources in rural areas; the exploitation of ecological and cultural functions of agriculture; the promotion of the agricultural value chain; and thus the integration of industries. The cultivation of new agricultural forms refers to introducing modern production and service concepts into agricultural and rural areas, transforming and upgrading traditional agricultural and rural areas, and building a more perfect rural–industrial system. The integration of agricultural service industries is a typical form of the integrated development of rural industries, which mainly introduces modern production factors into agriculture in a market-oriented way through service organizations and uses advanced varieties, technologies, and equipment to transform and upgrade traditional agriculture. The improvement of interest-linkage mechanisms can result in a community of interests between farmers and enterprises and can integrate farmers into the agricultural industry chain. Therefore, this study used the above indicators to build its rural–industrial integration index system, and it used the entropy method to measure the level of rural–industrial integration, as shown in Table 1.

**Table 1.** Evaluation index system of rural–industrial integration and the digital economy.

| Primary Index | Secondary Index | Three-Level Index | Index Calculation Method | Direction | Data Source |
|---|---|---|---|---|---|
| The level of integration of rural industries | Extension of the agricultural industry chain | Share of operating income in the agricultural product processing industry | Revenue of the agricultural product processing industry/gross output value of the primary industry | + | [83–88] |
| | Expansion of agricultural versatility | Development of leisure agriculture | Annual marketing revenue of leisure agriculture/gross output value of primary industry | + | |
| | Cultivation of new forms of agriculture | Proportion of agricultural land area in facilities | Total area of facility agriculture/cultivated area | + | |
| | Integration of agricultural services | Proportion of agriculture, forestry, animal husbandry, and fishery services | Total output value of agriculture, forestry, animal husbandry, and fishery services/gross output value of the primary industry | + | |
| | Improved interest-linkage mechanism | Number of farmers' cooperatives per million people | Number of specialized farmer cooperatives/rural population | + | |
| Primary index development level of the digital economy | Network popularization | Internet penetration | Number of Internet users/number of permanent residents | + | [33,89–93] |
| | General-purpose equipment | Mobile phone penetration | Number of mobile phones per 100 people | + | |
| | Digital penetration | Total telecommunications services per capita | Total volume of telecommunication services/number of permanent residents | + | |
| | Digital integration | Development of digital finance | Digital financial inclusion index | + | |
| | Input level | Number of Internet employees | People working in computer services and software/proportion of employees in the unit | + | |

The development level of the digital economy was selected as the explanatory variable. In terms of digital economy indicators, this study referred to the research of Huang [89], Ran [90], Wu [33], Kong [91], He [92], and Wu [93] and selected five indicators: network popularization, general-purpose equipment, digital penetration, digital integration, and

input level. Of these, network popularization and general equipment (which are necessary conditions for the generation of the digital economy) can be used to measure the basic environment and supporting conditions of the digital economy. Digital penetration and digital integration have shown that digital technology can widely penetrate all aspects of production, distribution, exchange, and consumption; the digital economy and the real economy are deeply integrated; and Internet practitioners are important subjects in the promotion of the development of the digital economy. Therefore, we chose the above indicators to build our digital economy evaluation index system, and we used the entropy method to measure the development level of the digital economy, as shown in Table 1. In the table, "+" indicates that the larger the index, the more conducive to the integration of rural industries.

Based on relevant studies, the following intermediary variables were selected in this model. (1) The per capita patent applications of various provinces in China were selected to measure the levels of scientific and technological innovation [94]. The digital economy provides a more open and inclusive environment for scientific and technological innovation, encourages all parties to share data and knowledge, accelerates the transformation and application of scientific and technological achievements, and penetrates the agricultural field through positive externalities, thus breaking the barriers between agriculture and the secondary and tertiary industries and promoting the integration of rural industries. (2) The per capita education years of rural residents in each province of China were selected to measure rural human capital [95]. In the process of the development of the digital economy, digital technology makes all kinds of learning resources easy to obtain. In addition to promoting basic knowledge on subjects such as Chinese and mathematics, digital technology also promotes the dissemination of agricultural knowledge in rural areas, improves rural human capital, and improves farmers' ability to apply digital resources and digital technologies, thus improving labor efficiency, optimizing the rural–industrial structure, and effectively promoting the integration of rural industries.

Based on relevant studies, the following control variables were selected in this model. The first was government financial support for agriculture, which effectively promotes rural social and economic development by improving the level of public services in rural society and promoting the good operation of agriculture, therefore impacting the integrated development of rural industries. The second was transportation infrastructure. Improvements in transportation infrastructure are conducive to promoting economic activities within and among rural areas as well as providing a good foundation for the integration and development of rural industries. The third was industrial structure. Generally speaking, the higher the proportion of the output value of the secondary and tertiary industries in the total output value, the more it can be used to promote the advanced development of the industry and promote the integrated development of rural industries.

Table 2 shows the symbols, basic meanings, and calculation methods of the explained variable, core explanatory variable, intermediate variable, and control variable. Table 3 provides descriptive statistics on the explained variables, explanatory variables, mediating variables, and control variables, including their symbol, sample size, mean value, standard deviation, minimum value, and maximum value.

**Table 2.** Variable names, symbols, basic meanings, and calculation methods.

| Variable Name | Symbol | Basic Meaning | Calculation Method | Data Source |
|---|---|---|---|---|
| Dependent variable | integration | Rural–Industrial Integration | Entropy method | As shown in Table 1 |
| Core explanatory variable | digital | Digital Economy | Entropy method | As shown in Table 1 |

**Table 2.** *Cont.*

| Variable Name | Symbol | Basic Meaning | Calculation Method | Data Source |
|---|---|---|---|---|
| Mechanism variable | innovate | Scientific and Technological Innovation Levels | The number of patent applications per capita is logarithmic | [94] |
| | Human | Rural Human Capital | Average years of schooling for rural residents | [95] |
| Control variable | government | Government Financial Support | Agricultural expenditure as a proportion of government expenditure | [40] |
| | traffic | Transportation Infrastructure | Regional highway mileage is logarithmic | [36] |
| | struct | Industrial Structure | The proportion of the total output value contributed by the secondary and tertiary industries in the overall value added to the total output value | |

**Table 3.** Descriptive statistics of variables.

| Variable Name | Symbol | Sample Size | Mean Value | Standard Deviation | Minimum Value | Maximum Value |
|---|---|---|---|---|---|---|
| Rural–Industrial Integration | integration | 330 | 0.150 | 0.108 | 0.014 | 0.535 |
| Digital Economy | digital | 330 | 0.225 | 0.164 | 0.013 | 0.958 |
| Scientific and Technological Innovation Levels | innovate | 330 | 10.105 | 1.439 | 6.219 | 13.473 |
| Rural Human Capital | Human | 330 | 7.843 | 0.626 | 5.925 | 10.160 |
| Government Financial Support | government | 330 | 0.115 | 0.033 | 0.041 | 0.204 |
| Transportation Infrastructure | traffic | 330 | 11.681 | 0.848 | 9.401 | 12.885 |
| Industrial Structure | struct | 330 | 0.901 | 0.053 | 0.742 | 0.997 |

Note: According to the data source of Section 3.1.1, we collected and summarized the data required for this article and obtained Table 3 through stata16.0 software.

### 3.2. Methods

Based on the regression model of Opler et al. [96], this paper established the following model to test the effect of the digital economy in promoting rural–industrial integration:

$$integration_{it} = \alpha_0 + \alpha_1 digital_{it} + \alpha_2 control_{it} + \varepsilon_{it} \tag{1}$$

In Formula (1), $integration_{it}$ represents the level of rural–industrial integration in region $i$ in year $t$, $digital_{it}$ represents the level of the digital economy in region $i$ in year $t$, $control_{it}$ represents the control variable, and $\varepsilon_{it}$ is the random disturbance term.

This paper drew on the intermediary effect model of Baron et al. [97] and adopted a step-by-step regression method to test the intermediary roles of scientific and technological innovation levels and rural human capital. The specific model is as follows:

$$inter_{it} = \beta_0 + \beta_1 digital_{it} + \beta_3 control_{it} + \varepsilon_{it} \tag{2}$$

$$integration_{it} = \gamma_0 + \gamma_1 digital_{it} + \gamma_2 inter_{it} + \gamma_3 control_{it} + \varepsilon_{it} \tag{3}$$

In Equations (2) and (3), *inter* is the intermediary variable, and the other variables have the same meaning as above.

In order to further analyze the impact of digital economy on the integration of rural industries in the region and surrounding areas, this study drew upon Dolores [98] to establish a spatial Durbin model to test the spatial spillover effect of digital economy. The specific model is as follows:

$$integration_{it} = \theta_0 + \theta_1 Wintegration_{it} + \theta_2 digital_{it} + \theta_3 Wdigital_{it} + \theta_4 control_{it} \\ + \theta_5 Wcontrol_{it} + \varepsilon_{it} \tag{4}$$

In Equation (4), $\theta_1$ represents the autoregressive coefficient, $W$ represents the spatial weight matrix, and the geographical adjacency matrix is used as the spatial weight matrix. If the two regions are geographically adjacent, the corresponding element in the weight matrix is given as 1 and otherwise 0; $Wintegration_{it}$, $Wdigital_{it}$, and $Wcontrol_{it}$ are the spatial lag terms of rural industrial integration, digital economy, and control variables, respectively. $\theta_3$ and $\theta_5$ represent the spatial interaction terms of digital economy and control variables, respectively.

The calculation of the model and formula in this paper was operated by stata16.0 software (StataCorp, College Station, TX, USA).

## 4. Results of Investigation

### 4.1. Results of Baseline Regression

The OLS model, fixed effects model, and random effects model were applied to regression Formula (1) to test the effect of the digital economy on promoting the integrated development of rural industries. The regression results are shown in Table 4: Model 1 indicated that without adding control variables, the regression coefficient of the digital economy was 0.344 and passed the test at the 1% significance level, which indicates that the digital economy can positively promote the integrated development of rural industries. Model 2 indicated that with the addition of control variables but without considering regional differences, the digital economy can still significantly promote the integrated development of rural industries in a positive way. The regression coefficient of the digital economy was calculated as 0.1655 and passed the test at the 1% significance level. The control variables were added to Formula (1), and the influence of regional differences was considered. The individual fixed effects model and individual random effects model were used to estimate the results. As shown in models 3 and 4, the regression results were consistent. The digital economic coefficient was 0.3037, which passed the test at the 1% significance level. This shows that the development of the digital economy can obviously promote the integration of rural industries. In summary, by observing the results of the four models, it can be seen that the digital economy has a positive promoting effect on the integrated development of rural industries, and Hypothesis 1 is verified. Furthermore, these results show that the level of government financial support for agriculture, the construction of transportation infrastructure, and the upgrading of industrial structure can promote the integrated development of rural industries.

**Table 4.** Results of baseline regression analysis.

| | Model 1: Fixed Effects | Model 2: OLS | Model 3: Fixed Effects | Model 4: Random Effects |
|---|---|---|---|---|
| *digital* | 0.3441 *** (0.0260) | 0.1655 *** (0.0429) | 0.3037 *** (0.0285) | 0.3037 *** (0.0285) |
| *government* | | −0.5308 *** (0.2000) | 0.7220 *** (0.1966) | 0.7219 *** (0.1966) |

**Table 4.** *Cont.*

| | Model 1:<br>Fixed Effects | Model 2:<br>OLS | Model 3:<br>Fixed Effects | Model 4:<br>Random Effects |
|---|---|---|---|---|
| *traffic* | | 0.0142 **<br>(0.0055) | 0.0112 ***<br>(0.0029) | 0.0112 ***<br>(0.0029) |
| *struct* | | 0.9367 ***<br>(0.1269) | 0.4381 *<br>(0.2451) | 0.4381 *<br>(0.2451) |
| Con | 0.0899 ***<br>(0.0051) | −0.8282 ***<br>(0.1443) | −0.3520<br>(0.2414) | −0.3520<br>(0.2414) |
| Area Control | Yes | No | Yes | No |
| N | 330 | 330 | 330 | 330 |
| $R^2$ | 0.3948 | 0.4627 | 0.5348 | 0.8858 |

Note: Standard error in parentheses. *, **, and *** indicate significance at the 10%, 5%, and 1% levels, respectively.

### 4.1.1. Endogeneity Test

In this paper, two methods were adopted to deal with the endogeneity problem. First, we referred to Meng [40] to delay the explanatory variables by one period, which can basically solve the reverse causality problem in a time sequence, and the results are shown in model 1 in Table 5: The digital economy with a lag of one phase still showed a significant positive impact on the integration of rural industries. In the GMM estimation of Nawaz [99], the results are shown in model 2 in Table 5: AR(1) was less than 0.1, and AR(2) was greater than 0.1, which obviously overcomes the endogenous problem. The results of the Hansen test showed that the model setting was reasonable.

**Table 5.** Endogeneity test results.

| | Model 1: Explanatory Variables Lag by One Stage | Model 2: System GMM |
|---|---|---|
| *digital* | 0.1563 ***<br>(0.0335) | 0.2663 **<br>(0.0983) |
| Control variable | Yes | Yes |
| Fixed effect | Yes | Yes |
| Sample size | 300 | 300 |
| $R^2$ | 0.888 | |
| AR(1) | | 0.015 |
| AR(2) | | 0.194 |
| HANSEN | | 0.586 |

Note: Standard error in parentheses. **, and *** indicate significance at the 5%, and 1% levels, respectively.

### 4.1.2. Robustness Test

In this paper, two methods were used to test the robustness, and the test results of the two methods are shown in Table 6: Model 1 represents the result of regression by eliminating samples. China has four municipalities directly under the central government. In order to avoid unobstructed factors caused by the institutional settings and government policies of municipalities directly under the central government, samples of these municipalities were removed, and the effect of the digital economy on rural–industrial integration was re-examined. Model 2 represents the regression results of the replacement method. The value range of rural–industrial fusion was 0~1, which meets the condition requirements of the constrained dependent variable model. The fixed effects Tobit model was used to replace the original model. The estimated coefficient of the digital economy was still significantly positive, which proved that the results of benchmark regression were robust and reliable.

**Table 6.** Robustness test results.

|  | Model 1:<br>Lagged by One Stage | Model 2:<br>Tobit |
|---|---|---|
| *digital* | 0.2596 ***<br>(0.0240) | 0.3037 ***<br>(0.0268) |
| *government* | 0.7120 ***<br>(0.1588) | 0.7220 ***<br>(0.1850) |
| *traffic* | 0.0079 ***<br>(0.0025) | 0.0112 ***<br>(0.0028) |
| *struct* | 0.4615 **<br>(0.1971) | 0.4381 **<br>(0.12308) |
| Con | −0.5130 ***<br>(0.1772) | −0.3520 ***<br>(0.2273) |
| Fixed effect | Yes | Yes |
| N | 290 | 330 |
| $R^2$ | 0.5166 | |

Note: Standard error in parentheses. **, and *** indicate significance at the 5%, and 1% levels, respectively.

*4.2. Results of Mediation Effect Test*

After analyzing the effect of the digital economy on promoting rural–industrial integration, this paper further explored the driving paths of the digital economy on rural–industrial integration, and the results are shown in Table 7: Column (1) is listed as the fixed effects benchmark regression result of the digital economy on rural–industrial integration. Columns (2) and (3), generated via the mediating effect testing of Formulas (2) and (3) regarding the levels of scientific and technological innovation, show that the digital economy has a promoting effect on the levels of scientific and technological innovation, improvements in scientific and technological innovation levels have a promoting effect on the integration of rural industries, and the coefficients were significantly positive at the level of 1%. Columns (4) and (5) also show the intermediary effect test results of Formulas (2) and (3) regarding rural human capital, showing that the digital economy has a promoting effect on rural human capital and that improving rural human capital has a promoting effect on rural–industrial integration; the coefficients were significantly positive at 1% and 5%, respectively. These results indicate that the levels of scientific and technological innovation and rural human capital are positive intermediary variables; that is, the digital economy can promote the integration of rural industries by promoting the levels of scientific and technological innovation and rural human capital. Hypotheses 2 and 3 are verified.

**Table 7.** Results of mediation effect test.

| Variable | (1)<br>*Integration* | (2)<br>*Innovate* | (3)<br>*Integration* | (4)<br>*Human* | (5)<br>*Integration* |
|---|---|---|---|---|---|
| *digital* | 0.3037 ***<br>(10.67) | 3.8596 ***<br>(0.2186) | 0.1812 ***<br>(0.0401) | 1.6766 ***<br>(0.1300) | 0.2569 ***<br>(0.0360) |
| *innovate* | | | 0.0317 ***<br>(0.0078) | | |
| *Human* | | | | | 0.0279 **<br>(0.0133) |
| Control variable | Yes | Yes | Yes | Yes | Yes |

**Table 7.** *Cont.*

| Variable | (1)<br>*Integration* | (2)<br>*Innovate* | (3)<br>*Integration* | (4)<br>*Human* | (5)<br>*Integration* |
|---|---|---|---|---|---|
| Fixed effect | Yes | Yes | Yes | Yes | Yes |
| N | 330 | 330 | 330 | 330 | 330 |
| $R^2$ | 0.3956 | 0.7227 | 0.4934 | 0.4837 | 0.4702 |

Note: Standard error in parentheses. **, and *** indicate significance at the 5%, and 1% levels, respectively.

### 4.3. Test Results of Spatial Spillover Effect

Due to digital technology, the digital economy has strong communication abilities; can break regional restrictions and speed up the circulation of resources, technologies, and elements; and may have an impact on the surrounding areas; that is, the digital economy can spatially affect the integration of rural industries in surrounding areas. Based on Dolores [98], this paper analyzed direct effects, indirect effects (spatial spillover effects), and total effects. Before the spatial spillover effect test, stata16.0 software was used to calculate the global Moran index under the spatial adjacency weight matrix, and the results are shown in Table 8: The *p*-values of the digital economy and rural–industrial integration were significant at the levels of 1%, 5%, and 10%, and Moran's I were all positive, indicating a positive spatial autocorrelation between the digital economy and rural–industrial integration in all regions; that is, there is a spatial agglomeration phenomenon of the digital economy and rural–industrial integration in China's provinces. Therefore, it was reasonable to use a spatial econometric model in this paper.

**Table 8.** The global Moran index of the integration of the digital economy and rural industries.

| Year | Digital Economy | | Digital Economy | |
|---|---|---|---|---|
| | **Moran's I** | **Z** | **Moran's I** | **Z** |
| 2011 | 0.127 ** | 1.755 | 0.491 *** | 4.550 |
| 2012 | 0.133 ** | 1.794 | 0.484 *** | 4.463 |
| 2013 | 0.098 * | 1.354 | 0.366 *** | 3.414 |
| 2014 | 0.089 * | 1.295 | 0.370 *** | 3.425 |
| 2015 | 0.112 * | 1.507 | 0.378 *** | 3.494 |
| 2016 | 0.147 ** | 1.819 | 0.375 *** | 3.550 |
| 2017 | 0.140 ** | 1.721 | 0.387 *** | 3.598 |
| 2018 | 0.135 ** | 1.627 | 0.312 *** | 3.010 |
| 2019 | 0.148 ** | 1.742 | 0.406 *** | 3.809 |
| 2020 | 0.162 ** | 1.889 | 0.414 *** | 3.842 |
| 2021 | 0.168 ** | 1.902 | 0.422 *** | 3.878 |

Note: Using data such as digital economy, rural industrial integration, and spatial matrix data, the Moran's index was calculated by stata16.0 software. *, **, and *** indicate significance at the 10%, 5%, and 1% levels, respectively.

The establishment of the spatial autocorrelation test indicated that it was reasonable to adopt a spatial econometric regression method. Before selecting the appropriate spatial measurement model, the LM test, LR test, and Hausman test were successively conducted. Based on the test results, the spatial Durbin model with fixed effects was selected. The results obtained by inputting the relevant data into Formula (4) are shown in Table 9: Column (1) contains the regression results of the spatial Durbin model. The digital economy coefficient was calculated as 0.2296, which was significant at the 1% level, indicating that the digital economy can significantly promote the integration of rural industries. According to our results, the digital economy promotes the integration of rural industries through three effects: the direct effect, the indirect effect of spatial spillover, and the total effect.

Column (2) contains the direct effect test results, showing that the local digital economy can significantly promote the integration and development of local rural industries. Column (3) contains the indirect effect test results, indicating that the development of the digital economy in surrounding areas can also significantly promote the integration of local rural industries; that is, there is a spatial spillover effect in the process of the digital economy promoting the integration and development of rural industries. Column (4) contains the total effect test results, that is, the total effect of the digital economy in the region and surrounding areas on the integration of local rural industries, and the total effects were calculated as positive.

**Table 9.** Regression results of the spatial Durbin model.

| Variable | (1) Spatial Durbin Model | (2) Direct Effect | (3) Indirect Effect | (4) Total Effect |
|---|---|---|---|---|
| *digital* | 0.2296 *** (0.0783) | 0.3439 *** (0.0604) | 1.0881 *** (0.2232) | 1.4320 *** (0.2555) |
| *government* | −0.0504 (0.1785) | −0.7903 *** (0.1861) | 1.1355 *** (0.4053) | 0.3453 (0.4197) |
| *traffic* | 0.0057 (0.110035) | −0.0020 (0.0072) | −0.0329 (0.0261) | −0.0349 (0.0316) |
| *struct* | −0.2670 (0.2303) | 0.3377 *** (0.1124) | 0.5092 (0.3565) | 0.8469 ** (0.3546) |
| Log-likelihood | 385.7476 | | | |
| $R^2$ | 0.5301 | 0.291 | 0.291 | 0.291 |
| ρ | 0.4738 *** (0.0705) | 0.374 *** (0.0735) | | |
| sigma2_e | 0.0042 *** (0.0004) | 0.004 *** (0.0003) | | |
| N | 330 | 330 | 330 | 330 |

Note: Standard error in parentheses. **, and *** indicate significance at the 5%, and 1% levels, respectively.

## 5. Discussion

The rapid development of China's digital economy has had profound impacts on social change and may also affect the process of industrial integration in rural China, and this study analyzed how China's digital economy affects rural–industrial integration. Our results are as follows. First, we found a significant positive relationship between the development level of the digital economy and the level of rural–industrial integration, and the development of the digital economy was found to promote the process of rural–industrial integration. In his study of Iranian rural areas, Zabih [31] believed that in the context of intelligent tourism, the use of digital technology is an important step in improving the efficiency and effectiveness of tourism. By integrating intelligent tourism into rural areas, it is possible to completely change tourism and stimulate economic growth in these areas. Rosalina [100] also found that there has been a growing body of research that emphasizes the importance of digitalization in rural tourism. Although previous studies have found that the digital economy is an important factor in promoting the integration of rural industries, there is a lack of empirical analysis to verify this finding. On the basis of describing how the digital economy promotes the integration of rural industries, this paper conducted an economic model test, again supporting the view that the digital economy promotes the integration of rural industries.

Second, there may be many ways for the digital economy to promote the integration of rural industries. This paper argues that the levels of scientific and technological innovation and rural human capital are important intermediary channels. Vial [101] found that digital transformation (DT) can be defined as a disruptive process where organizations

change value-creating processes by adopting digital technologies in response to changes in the business environment. Mayakova [102] believed that digital transformation boosts innovation, as it requires the acquisition of new knowledge and skills, calls for new forms of collaboration within the organizations and across industries, promotes the creation of new business models, and leads to the sustainable usage of organizational resources. In the study of India's digital agriculture architecture, Acharya [103] believed that some technical measures such as AI algorithms provide great benefits and applicability in agriculture and may turn agricultural supply chain management challenges into opportunities. Zambon [104] believed that with the continuous development of innovation and technology, the virtualization of agro-food supply chains in partnership with stakeholders such as farmers, wholesalers, and retailers will truly revolutionize the agriculture sector. Olga [105] found that with the development of the digital economy, digital technology has a positive effect on improving human capital in the Russian Federation. Alharthi [106] found that financial inclusion can improve the human capital of sub-Saharan African (SSA) countries. Daniel [107] believed that human capital is an important resource in the process of agricultural modernization transformation in Africa. Zhang's [108] study found that rural human capital has a promoting effect on the integration and development of rural industries. These scholars analyzed the relationship between scientific and technological innovation, rural human capital, the digital economy, and rural–industrial integration, but they did not link them together for research. This study analyzed how the digital economy promotes rural–industrial integration and found that the digital economy can promote rural–industrial integration by improving the levels of scientific and technological innovation and rural human capital.

Third, by studying the impact of Thailand's digital divide on short-term transport policy, Hironori [109] found that the level of technological development in neighboring regions is often similar, and geographical constraints may widen the digital divide. Philipp [110] took European countries as the research object and found that there is spatial correlation in the economic growth of European countries. Wolfgang [111] studied German counties and found that broadband deployment in German counties induces not only substantial economic benefits in terms of direct effects within counties but also positive regional externalities across counties. At present, there is a lack of spatial effect analysis of the digital economy on rural–industrial integration. Using a spatial econometric model, this paper found that there is spatial spillover effect in the process of the digital economy promoting rural–industrial integration, which enriches relevant research fields and is conducive to formulating policies to promote the coordinated development of the digital economy and rural–industrial integration among regions.

The limitations of this paper are as follows. In terms of selecting evaluation indicators of digital economy and rural industrial integration, this study referred to international and domestic excellent journal papers and selected as many indicators as possible that are recognized and mature by most scholars. However, in view of the data availability and complexity of digital economy and rural industrial integration, the indicators selected may have limitations. Second, this paper analyzed how the digital economy promotes the integration of rural industries. In practice, there may be multiple channels in the process of promoting the integration of rural industries with the digital economy, but this paper only selected the levels of scientific and technological innovation and rural human capital as the intermediary variables, and it did not analyze and verify other possible intermediary variables. Third, this paper drew on relevant studies and chose a regression model to study the problem. However, due to space limitations, the regression model was not compared with other models, and no more consideration was given to which research method was more appropriate and low-cost, which may have affected the rigor of this study. Fourth, the data in this study came from China's statistical databases, and the model's research object comprised various Chinese provinces, so our conclusions and suggestions may be limited to China. Can the analytical ideas and research methods of this paper be applied to other countries or regions? The authors of this paper argue that the development of the digital

economy is a worldwide trend and will bring many social and economic benefits. As a part of social development, rural–industrial integration will result in dividends brought by the development of the digital economy. Therefore, if other countries or regions have developed their digital economies and can collect the indicator data on the integration of the digital economy and rural industry, then the model in this paper can still be applied to analyze how the development of local digital economies promotes the integration of local and rural industries according to local conditions.

## 6. Conclusions

This study analyzed how China's digital economy promotes rural–industrial integration and verified its findings using panel data and regression models of 30 provinces (municipalities and districts) in China from 2011 to 2021. Our results are as follows. First, the development of the digital economy can significantly promote the integration of rural industries. Second, in the process of promoting the integration of rural industries with the digital economy, the levels of scientific and technological innovation and rural human capital are two positive intermediary variables. Third, there is a spatial spillover effect in the process of promoting the integration of rural industries with the digital economy; that is, the development of the digital economy in a region will promote the integration of rural industries in the region and surrounding areas, and the development of the digital economy in the surrounding areas will also promote the integration of rural industries in the region.

In order to actively guide the development of the digital economy, we must find and use the best paths and channels to promote the integration of rural industries. We propose the following tasks. First, we should increase investments in digital rural construction and infrastructure construction; actively publicize the development advantages of the digital economy; use the digital economy to open up the upstream and downstream industrial chains of agriculture; guide agricultural production, processing, transportation, sales, and other processes based on local resource endowments; and stimulate the vigorous rise of rural tourism, leisure agriculture, the residential economy, and other new forms of rural business. We should also continue to promote the deep integration of rural industries. Second, we should create a conducive atmosphere for scientific and technological innovation and pay attention to the training of scientific and technological talent; apply scientific and technological innovation to the development of rural industries; promote the application of cloud computing, Internet of Things, and artificial intelligence in the integration of rural industries; strive to improve the education level of rural residents; and improve the Internet knowledge and digital economy literacy of agriculture-related personnel. We should furthermore cultivate modern high-quality agricultural and rural digital talent and give full play to the positive intermediary roles of scientific and technological innovation levels and rural human capital in the process of the digital economy promoting rural–industrial integration. Third, the digital economy has a spatial spillover effect on the promotion of rural–industrial integration, which means that there is a connection between the digital economy and rural–industrial integration among various regions. Therefore, the government needs to take the overall situation into consideration, improve its top-level design, formulate good policies to guide the development of the digital economy and rural–industrial integration, actively encourage mutual learning and enhanced communication between various regions in the integration of the digital economy and rural industry, rationally plan the coordinated construction of digital infrastructure between regions, and provide guarantees for the smooth flow of capital and technology in various regions. We should encourage complementary regional agricultural information and resource sharing; give full play to the spillover effect of the digital economy to promote the integration of rural industries; promote the effective connection of data between different regions, departments, and enterprises; and realize the connectivity and sharing of data resources between different platforms to provide strong support for the process of promoting the digital economy to promote the integration of rural industries.

**Author Contributions:** Conceptualization, C.S. and Z.Z.; methodology, C.S.; software, Z.Z.; validation, Z.Z.; formal analysis, Z.Z. and J.W.; investigation, C.S. and J.W.; data curation, Z.Z.; writing—original draft preparation, Z.Z.; writing—review and editing, C.S. and J.W. All authors have read and agreed to the published version of the manuscript.

**Funding:** This research was funded by the Ministry of Agriculture of People's Republic of China through the China Agriculture Research System, grant number CARS-48.

**Institutional Review Board Statement:** Not applicable.

**Informed Consent Statement:** Not applicable.

**Data Availability Statement:** The data in this paper are publicly available and mainly came from the National Bureau of Statistics of China: http://www.stats.gov.cn/sj/ (accessed on 5 April 2023).

**Conflicts of Interest:** The authors declare no conflict of interest.

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
