# Peer review of "How Can the Digital Economy Promote the Integration of Rural Industries—Taking China as an Example"

_agriculture, doi:10.3390/agriculture13102023_

Round 1

Reviewer 1 Report

Comments and Suggestions for Authors

The paper has interesting title and is related with modern scientific problematic, but at the moment in my opinion after detailed reading, it is more still a draft version than ready for publishing material.

In my opinion, the title is not properly related to the content. I also see a big problem with the current structure of the paper, the unclear research goal related to the title and content, the selection and review of scientific literature, the discussion of the results and the conclusions drawn. Therefore, my review is negative and I do not recommend this paper in its current form for publication.

Detailed comments:

1) The title. First of all, this is a text only related to China, and does not concern the situation in other countries or continents, so the title should include something like "on the example of China" or "in Chinese conditions". Secondly, the content of the paper is more about answering "if", not "how" the digital economy promotes the integration of the agricultural industry.

2) The abstract should have a clearly defined scientific purpose.

3) The structure of the paper. Some parts of the text should be included in other chapters. There are also too many subchapters (this especially applies to the current chapter 4, because I see no point in making subchapters with a maximum of 2-3 sentences, but in other cases it is also worth considering whether it is possible to combine some subchapters).

4) Introduction. In my opinion, the research questions are inappropriate because the research conducted shows whether there is a relationship, but not what its nature is or what exactly it changes in the agricultural industry. Therefore, they should be changed and the research goal should be clearly specified and consistent with research based on empirical data. Or maybe, based on the last sentence (lines 52-53), the goal is to determine what path of development the digital economy should have so that it supports the strategy of revitalization of rural areas? However, the paper does not contain any information about the assumptions of this strategy, nor any statistical research in this area.

In general, the content of the introduction suggests the relationship between China's development model and its impact on the agricultural industry, and the digital economy is only one of the elements of this model and development conditions.

5) Chapter 2. I don't see a definition of this sector in the subsection on the digital economy... There are various aspects of research on the digital economy, but there is no information on how we define the digital economy.

The authors refer to research from the perspective of Marx's political economy (line 66), but aren't there studies of what this looks like in economics as a whole? I think that promoting Marx's ideas is of very questionable importance due to the long-term ineffectiveness of the economic system based on his assumptions and the strong influence of the ideology on the economy and society, which has resulted in great human sacrifices throughout history. Line 81. The authors claim that the concept of integration of the agricultural industry appeared in the 1990s. Perhaps on the Chinese market, however, in general, issues related to the integration of the agricultural sector, cooperatives theoretical issues related with a spatial aspects of management theory and development of rural areas are much older. Line 87. What is the difference between the concepts of "agricultural income" and "farmers' income"? They have the same meaning, unless it concerns the entire sector and individual farms, but this is currently unknown. Line 92. The authors try to describe three aspects of agricultural industry integration, but the second one is called "mechanism" instead of the word "aspect". Are you sure this sentence is spelled correctly? Line 93-96. The sentence is long and unstylish. I guess something is missing there, because it describes research suggestions regarding "digital", but it is not clear what. Perhaps that is why the beginning of the sentence is not consistent with the end of the sentence in this version? Moreover, this entire sentence is not coherent with the introductory sentence earlier (line 92). Line 98, There is a word "county". Is it supposed to be like that or is it a typo? Lines 115-128. In my opinion, this fragment should be moved to the introduction or methodological chapter, or to the summary. This is not a fragment of the art of the stuy, but only the authors' own opinions (there are no citations of external items), although it is included in the "Theoretical background" chapter.   6) Chapter 3. Large fragments of the text are actually a continuation of general "theoretical background" information and should be included there. However, research hypotheses with a short explanation of why they are formulated this way, in addition to presenting research methods, should be part of the Methods and materials chapter. Overall, the hypotheses sound very similar, so in my opinion they can be combined and their number reduced. Generally, the name of the entire chapter does not fully correspond to the content, and the titles of the subchapters in particular are questionable. For example, the impact of the digital economy on the integration of the agricultural industry - there are no measures determining this impact, only some opinions on the ways of impact without providing hard data. Lines 132-158. There are no external citations at all. If this is the authors' own work, where is the data to support their opinion? Lines 133-136. Factors such as knowledge and technology, labor and capital have always been introduced into the agricultural sector. The meaning of the sentence suggests that they did not exist before and that digital technology is only introducing it. Lines 178-181. What is the source of the information?   7) Chapter 4. The title should be simplified to "Materials and methods". The content should include all methodological issues, sources and research hypotheses, and the number of subchapters should be reduced or no subchapters at all. In addition, all tables should have a short commentary discussing the most important results presented in tables 1-4. 8) Chapter 5. The title should be simplified to something like that "Results of investigation". I think that the current division into subchapters is unnecessary.   9) Chapter 6. I have some doubts about the proposed conclusions and suggestions. Perhaps they result from translation problems and this makes it more difficult to clearly understand the meaning of the authors' opinions. Lines 415-417. Infrastructure development itself has the goal of improving this infrastructure and even introducing new elements of such infrastructure. So I don't know why this sentence mentions "special consideration"? Lines 418-419. Aren't digital technology and digital finance part of the digital economy? If so, why are they listed as separate spheres? Lines 421-422. Is this a well-translated sentence? And isn't it more about the "advantages" than the "characteristics" of the digital economy? Moreover, merely emphasizing the "characteristics" of the digital economy is unlikely to promote the integrated development of rural industry Lines 432-434. Where did this conclusion come from? The content of the paper does not include detailed research on demonstration parks and clusters and its impact to the rural development. Lines 434-436. Is this correct form and content of the sentence? I have doubts about it.   10) References. The number of items is adequate, but they are all written only by Chinese researchers. You want to publish the article in an international journal, so the theoretical background or discussion of the results should also include items from other parts of the world. Also those that have a critical view of a given issue, and not only those that show the positive aspects of phenomena related to the selected research topic. A scientist's work is a search for truth and discussion with other points of view. I think that in the resources of MDPI journals or other publishers you can find interesting items on this topic.
I also have doubts about the functionality of links with DOI numbers and the proper description of the titles of works already included in the references.   11) In my opinion, between the research results and the conclusions, there should be a chapter on the discussion of the results. This should be based on research on similar issues by other authors (taking into account my comments on the bibliography used).    

    ​Sprawdź szczegóły   Comments on the Quality of English Language

In my opinion, this version of the paper should be checked again and corrected in the correct English translation.

Examples:

Using similar words or unnecessary repetitions (line 87 - agricultural income and farmers income; lines 120 and 122 - second, secondly).

Lines 93-96. There is a problem with the form, style and proper meaning of the entire sentence

Line 98. I have doubt about using the word "county".

Line 207. The word "iteration" - what does it mean?

Line 415. Use a capital letter after a colon.

Lines 434-436. There is a problem with the form, style and proper meaning of the entire sentence.

Author Response

Dear Editor and Reviewers,

Manuscript Number: agriculture-2586045

Title: How can the digital economy promote the integration of rural industries

Thank you for your comments concerning our manuscript. Those comments are all valuable and very helpful for revising and improving our paper, as well as the important guiding significance to our researches. We are very sorry for some negligence and error that we made in this manuscript. We have carefully studied the comments and made significant revisions to the full text and chapters based on the valuable suggestions of the reviewers, as well as polishing the paper language at the DMPI professional institute. Now, we assure the reviewers that everything has been corrected in the revised manuscript.

We hope it meets with approval.

Please see the attachment for the main corrections of the paper and the point-by-point responses to your comments

Reviewer 2 Report

Comments and Suggestions for Authors

Please read the attached document, it might give you some useful ideas.

Comments on the Quality of English Language

Please make sure you read the paper carefully again. There are several sentences that must be improved for coherence or clarity. For example:

Original: "The research on rural industry integration mainly focuses on three aspects. First, the primary focus of research is rural industry integration."

Revised: "Studies on rural industry integration predominantly focus on three aspects. First, the pivotal area of exploration concerns rural industry integration."

i.e. do not repeat the word research twice or even more in the paragraph and make the sentences more concise. 

Author Response

(The authors gave the same response as above.)

Reviewer 3 Report

Comments and Suggestions for Authors

The authors stated an interesting title for the work, but its content has several defects.

1) The list of sources contains the works of authors from Asia only. This may lead to biased conclusions, since the view of the problem may be one-sided.

2) The authors consider the digital economy as an abstract entity. Attention should be paid to specific digital technologies, to explain what is their mechanism of influence on the rural industry, what is the scale of influence. Or maybe not the digital economy affects the development of the rural industry, but scientific and technological progress in general or other factors?

3) The advantages of regression analysis are known. The authors should also consider other possible ways to solve the problem, comparing regression analysis with other methods and justifying the choice. It is possible that the problem can be solved by less costly methods. The authors should think about this and consider this issue in the article.

4) The authors should discuss the possibility of applying the model to other territories.

5) The authors did not provide a theoretical basis for building the model. Why did they choose these variables? Why do these variables correspond to the concepts of the model? Why were these model concepts chosen?

Author Response

(The authors gave the same response as above.)

Round 2

Reviewer 1 Report

Comments and Suggestions for Authors

I would like to thank the authors for the corrections made. However, I still have reservations about the methodological part and the bibliography. Below are my comments.

1) The methodological chapter is incomplete and has an incorrect structure. There is nothing about materials in the "Materials" subsection, it is all about hypotheses! Subsection 2.2.1 "Data and variables" is part of the "Methods" subsection. This is incorrect because according to methodological canons, data are not methods! However, it does not list all the methods used to create the article, which are, for example, briefly mentioned when describing the research results. In my opinion, the methodological chapter should list all methods and briefly describe or indicate literature items that describe these methods in detail. That's why we have a chapter on methodology in scientific papers.

2) In my opinion, each figure and table should have a description of the source - whether it is the authors' own research or developed on the basis of other research.

3) Page 9, line 1475 - are these few words a description of the source for the table, or part of the text below? Please check and correct it.

4) The number of items in the bibliography list is very large, but some items are missing authors, and the citation style throughout the list contains errors in relation to the MDPI editorial requirements. Moreover, citations to authors from outside China are sparse. It would be particularly useful to have a broader perspective from other countries or continents in the "Discussion of results" chapter on the digital economy and its connections with the agricultural sector. In my opinion, this is the imperfection of this version of the paper.

5) Bibliography, position no. 34. The authors of item no. 34 refer to the Japanese school of Nara Imamura, but it is an article written by a Chinese author. Are there really no original sources straight from Japan on this subject?

Author Response

Dear Editor and Reviewers,

Manuscript Number: agriculture-2586045

Title: How can the digital economy promote the integration of rural industriesTaking China as an example

Thank you for your comments concerning our manuscript. Those comments are all valuable and very helpful for revising and improving our paper, as well as the important guiding significance to our researches. We are very sorry for some negligence and error that we made in this manuscript. We have carefully studied the comments and made revisions to the text and chapters based on the valuable suggestions of the reviewers, we assure the reviewers that everything has been corrected in the revised manuscript.

We hope it meets with approval.

Please refer to the attachment for details.

Reviewer 3 Report

Comments and Suggestions for Authors

The authors have made changes. The article has improved in quality. But two issues have not been fully resolved.

1. The authors attempted to explain how the digital economy affects the rural industries. But their theses are declarative. The authors should not just indicate this or that digital technology as the cause, but as a consequence how good it will be for the rural industries. Authors must show the mechanism or way to achieve the effect.

2. The authors should discuss in more detail the problem of the lack of generally accepted indices. It is necessary to consider specific examples and prove the narrowness of this methodology. Authors should include one more section in the work.

Author Response

(The authors gave the same response as above.)
